# The Application of Terahertz Technology in Corneas and Corneal Diseases: A Systematic Review

**DOI:** 10.3390/bioengineering12010045

**Published:** 2025-01-08

**Authors:** Bing Jie Chow, Chang Liu, Mingyi Yu, Isabelle Xin Yu Lee, Jodhbir S. Mehta, Qing Yang Steve Wu, Regina Wong Kay Ting, Ke Lin, Yu-Chi Liu

**Affiliations:** 1Barts and the London School of Medicine and Dentistry, Queen Mary University of London, London E1 2AD, UK; 2Tissue Engineering and Cell Therapy Group, Singapore Eye Research Institute, Singapore 169856, Singapore; 3Department of Cornea and External Eye Disease, Singapore National Eye Centre, Singapore 168751, Singapore; 4Ophthalmology and Visual Sciences Academic Clinical Program, Duke-NUS Medical School, Singapore 169857, Singapore; 5Institute of Materials Research and Engineering, Agency for Science, Technology and Research, Singapore 138634, Singapore

**Keywords:** terahertz, corneas, corneal hydration, corneal scars, dry eye

## Abstract

Terahertz (THz) waves reside in the electromagnetic spectrum between the microwave and infrared bands. In recent decades, THz technology has demonstrated its potential for biomedical applications. With the highly unique characteristics of THz waves, such as the high sensitivity to water and optimal spatial resolution coupled with the characteristics of the human cornea, such as its high water content, THz technology has been explored as a potential modality to assess corneas and corneal diseases. This systematic review provides an overview of the characteristics of THz waves, the safety profile of THz technology in the field of ophthalmology, and its clinical applications, including the objective evaluation of the corneal hydration, tear film, dry eye disease, corneal endothelium, corneal elasticity, and scarring. The paper also presents our viewpoint on the present challenges and future directions of THz technology prior to its broader integration into clinical practice.

## 1. Introduction

Terahertz (THz, 1012 Hz) radiation constitutes the band of electromagnetic waves with a frequency that lies between the infrared (IR) and microwave regions in the electromagnetic spectrum, predominantly in the 0.1–10-THz region (Figure 1) [1]. Known alternatively as “far-infrared”, which was commonly used before the 21st century, the terahertz band was termed the “terahertz gap” due to enduring obstacles in creating and detecting these frequencies effectively. However, since then, research in the generation, detection, and useful application of THz technology has advanced considerably in the past two decades, with an increasing number of applications emerging across various disciplines such as materials science, physics, communications, security, and biomedicine [2]. With corresponding wavelengths ranging from 3 to 0.033 mm, the THz radiation band exhibits a unique synthesis of the high spatial resolution associated with the far infrared light, with reduced scattering and large interaction with hydrogen bonding in bulk water characteristics of the millimeter-wave radiation. The unique property of THz radiation renders it a potentially valuable tool to evaluate water content in target materials, promoting the application of THz radiation in spectroscopy and imaging technology in the realm of biomedicine.

The initial exploration of THz radiation as a modality for imaging of biological systems dates back to 1995 when Hu et al. [3] published transmission images of a leaf immediately removed from a host plant and 48 h post-drying, proposing it for wet tissue imaging, and confirmed its diagnostic potential for biological tissues. With the advent of advancing field of terahertz biophotonics, the potential application of THz radiation in spectroscopy and imaging technology within the context of biomedical settings has been garnering substantial interest. This is attributed to THz radiation’s unique properties well-suited to biomedical research, primarily its ability to excite low-frequency molecular vibrations, including hydrogen bonds, van der Waals, or other non-bonded interactions. When THz radiation propagates through biomolecules, each individual biomolecule generates its distinctive spectral vibrational signature in the THz range. The generated “THz fingerprint” could subsequently be utilized for identifying and characterizing objects based on THz spectroscopic measurements [4]. Consequently, the high dielectric constant and absorptive characteristics of water for THz radiation indicate an increased sensitivity of THz radiation to water content in physiological tissues, which can be effectively harnessed in biomedical imaging [5]. In addition to its high degree of water content sensitivity that could be achieved, the application of THz imaging techniques has also attracted significant interest due to its non-invasive, non-destructive, non-ionizing, and sensitive nature. This is also particularly relevant in light of the limitations of the current transmission-mode medical sensing methods, which are hindered by the high attenuation exhibited in most biological tissues [6,7]. THz imaging has thus shown significant advantages in situ for distinguishing between normal and pathological tissues. However, owing to the high absorbance of water within the THz frequency region, it remains pertinent to recognize that THz waves possess a limited penetration depth into wet biological tissues, up to a few hundred micrometers in the superficial skin surface [8]. Consequently, various studies have been undertaken to explore the potential of THz scanning in analyzing a broad spectrum of body tissues and pathologies, including the ocular surface and cornea. In this systematic review, we aim to explore the viability of this promising technology for the cornea and ocular surface pathologies.

## 2. Method of Literature Search

This systematic review was conducted in accordance with the standards of the Preferred Reporting Items for Systematic Reviews and Meta-Analyses (PRISMA) with the primary research aim of examining the potential applications of THz technology within the cornea. The query performed in PubMed was a combination of the following search terms: “(Terahertz OR Terahertz Spectroscopy OR Terahertz Imaging OR Terahertz OR Terahertz Radiation) AND (Cornea OR Ocular Surface)”.

Databases were searched from inception to 22 October 2024, with no filters applied. From this search in the PubMed database, 32 references, published from 2008 to 2023, were retrieved. Articles of all study types (clinical trials, meta-analysis, randomized controlled trials, review, and systematic review) pertaining to the application of THz technology on the cornea from the database search were included in this article with duplicates eliminated. The exclusion criteria include articles not written in the English language or those without full text. Following the application of inclusion and exclusion criteria and a relevance screening of titles and abstracts, 17 references were identified as suitable for further review. The full-text versions of these articles were then assessed for eligibility. Relevant studies and publications identified through cross-referencing within the cited sources were also incorporated into the reference list to provide a more comprehensive discussion of THz technology applications in other body systems, as well as the anatomy, physiology, and function of the cornea. A total of 70 articles were included in the final manuscript. The methodology for reference selection is illustrated in Figure 2.

## 3. The Application of Terahertz in Medical Fields

Hydration-related variations in dielectric function have been measured in different tissue types and evaluated in comparison between diseased and healthy tissues. This includes the incorporation of THz technology for in vivo imaging of carcinomas, melanoma, skin burns, inflammation, scarring, and diabetic foot ulcers [9,10,11,12,13].

Cancer has been a major global cause of mortality. Early diagnosis, prompt treatment, and consistent monitoring play a crucial role in significantly reducing cancer-related mortality rates. The prospects of cancer imaging across various body tissues with THz imaging have been investigated by several research groups [9,12,14,15,16,17]. Within the context of liver cancer, THz spectroscopic imaging successfully distinguished cancerous areas in paraffin-embedded liver cancer samples by analyzing variations in refractive index and absorption gradient [14]. In a similar vein, the application of THz transmission-type spectroscopy revealed a higher refractive index in cancerous breast tissue upon imaging of breast cancer specimens [15]. In an ex vivo setting, the THz pulsed imaging system has also been shown to differentiate cancerous and healthy breast tissues with a reported accuracy of 92% [16]. Nonetheless, the deployment of THz spectroscopy in clinical settings remains under debate due to challenges such as penetration depth and limited evaluation of tissue cellularity at a macroscopic level, rather than a microscopic one, which hampers the acquisition of information [18]. These challenges remain active areas of research.

Among the various applications, skin cancer imaging presents as one of the most prominent areas of THz medical imaging research. This can be partially ascribed to skin cancer’s manifestation on the body’s surface, as it circumvents the constraint of the limited penetration depth inherent in THz imaging. Previous ex vivo studies have reported the successful identification of basal cell carcinomas (BCCs) of the human skin via THz imaging [9,17]. Additionally, the first in vivo imaging study using THz imaging on five patients with BCC revealed a significant broadening of the THz pulse reflectivity in affected regions compared to healthy skin tissues. The study validated the potential of THz imaging to distinguish between diseased and healthy skin regions, highlighting its efficacy in delineating BCC tumor margins [12].

In addition to skin cancers, the application of THz imaging has also been investigated for skin burn wound assessment. The ability to differentiate between superficial and deep partial-thickness burns is fundamental to the effective assessment of burn wounds, as this distinction informs the decision-making process regarding the need for surgical interventions, such as burn wound excision [19]. However, existing methods primarily rely on visual and tactile assessment, which can be prone to subjectivity and inaccuracy, thereby highlighting the value of an objective tool to enhance assessment accuracy [20]. In vivo burn studies have evaluated the capability of THz imaging by successfully imaging the partial and full-thickness burns on Sprague Dawley rats through THz reflectivity [21]. The animals underwent THz imaging and were subsequently compared with histopathological analysis. High-resolution THz imaging of the skin burns revealed a significantly increased THz reflectivity in severely damaged skin regions, in contrast to the less affected surrounding tissues. This is thought to reflect an increase in local water content within the damaged regions, driven by coagulation, which causes the formation of edematous tissue [22]. This thereby illustrates a proportional correlation between THz reflectivity and the severity of the burn wounds [21,22].

## 4. The Application of Terahertz in Corneas

The cornea is the eye’s outermost structure and is a transparent avascular tissue that, in conjunction with the lens, supplies the eye’s optical power to enable the convergence of incoming light onto the retina [23]. Additionally, the cornea performs vital roles such as maintaining ocular integrity and acting as a protective barrier against environmental factors [24,25,26]. With THz technology proving successful in multiple fields of medicine, particularly in relation to hydration-related conditions, there is growing interest in its potential for the objective assessment of corneal health. Corneal transparency is critical to the cornea’s function of providing refractive power but can be compromised by pathological processes including corneal scars or corneal edema, with the latter directly linked to variations in corneal tissue water content (CTWC) [27]. Comprising 78% water by volume [28], the cornea’s high water content, the homogeneity of its stromal sublayer, and its relatively stable physiological characteristics compared to other body structures present THz imaging as a promising approach for evaluating various aspects of corneal health, especially given the high sensitivity of THz waves to water [29].

### 4.1. THz Scanning System Used in Corneal Scans

The illustration of the THz scanning system employed in the authors’ work is depicted in Figure 3 [30,31,32,33,34]. The system operates in reflection mode, characterized by a reflection angle of approximately 45 degrees between the incident and reflected THz waves. Upon reaching the cornea, the THz wave interacts with the tissue, producing a reflected wave that encodes critical information regarding the corneal structure and composition. The detector subsequently collects this reflected wave, enabling a comprehensive analysis of the cornea’s properties. This setup is instrumental in advancing diagnostic methodologies and research in ophthalmic applications, leveraging the unique capabilities of THz spectroscopy.

### 4.2. Safety Profile of THz Radiation on Corneas

Beyond its non-invasive and non-destructive nature attributes, THz radiation is non-ionizing and defined as electromagnetic radiation that lacks the photon energy required to ionize atoms or molecules. This contrasts with traditional imaging methods such as X-ray scanning or magnetic resonance imaging, which are ionizing and pose inherent radiation risks, thereby highlighting the THz scanning systems’ critical advantage in being biologically innocuous.

The safety profile of THz radiation on corneas has also been investigated and tested across several studies [30,35,36]. On a cellular level, an assessment of human corneal epithelial cells from the eye exposed to 0.12 THz radiation for 24 h indicated no significant effects on the micronucleus frequency, morphology, or heat-shock protein (HSP) expression of the cells [35].

An ex vivo study, which utilized the use of THz radiation between 0.5 and 20 THz at a power of 80 uW in a porcine cornea for 4 h, reported no discernible degradation in the corneal structure through attenuated total reflection spectroscopy [36]. In an in vivo rabbit model by Liu et al. [30], excessive exposure to THz radiation at 0.3 THz at a power of 40 uW to 8 rabbits (16 eyes) was applied for either 1 h or 4 h. The eyes were subsequently analyzed on a tissue, cellular, structural, and functional level after THz exposure, with the rabbit eyes categorized into three groups, a 1-h THz exposure group, a 4-h exposure group, and a 4-h exposure group, and analyzed a week post-exposure to assess for delayed tissue responses. No corneal haze, lens opacification, or thermal effect was observed in all the corneas exposed to varying durations of THz exposure upon clinical assessment. Additionally, histological and immunochemistry analysis revealed no abnormalities including inflammation, fibrosis, or ischemia in the corneas, retinas, and lens. Quantitative real-time reverse transcription polymerase chain reaction also revealed no significant increase in mRNA expression of proteins such as HSP90AB1, DNA damage-inducible transcript 3, and early growth response-1, which serve as markers of heat attenuation, cell stress response, and a transcriptional regulator, respectively. In vivo, confocal microscopy (IVCM) and electroretinography also yielded no abnormalities in terms of the stromal keratocytes in the cornea and retina function, respectively. These findings suggest that THz radiation has good biological safety profiles for the eye, laying the groundwork for further research into the application of THz imaging systems in ophthalmology.

### 4.3. The Application of THz Technology in the Quantification of Corneal Sublayers

Ke et al. investigated human cadaveric and rabbit corneas using THz broadband spectroscopy and reported the potential role of THz spectroscopy in elucidating non-invasive characterization of the corneal sublayers [31,32]. By altering the position of the THz receiver from direct reflection along the corneal surface, the adjustment enabled the visualization of individual corneal sublayers, such as the epithelium layer and the stromal layer at different distances of deviation.

### 4.4. THz Imaging in Dry Eye Disease

Dry eye disease (DED) is one of the most prevalent ocular surface diseases affecting one-third of the global population, with its incidence increasing with advancing age [37,38]. As a chronic multi-factorial disease characterized by symptoms such as tear film instability and ocular discomfort, DED can arise from a myriad of factors, including ocular surface inflammation and dysfunction of the meibomian gland, lacrimal gland, or corneal nerves [39]. Conventional diagnostic methods for DED, such as fluorescein tear break-up time and Schirmer test, are constrained by issues of poor reliability and reproducibility, posing significant challenges in both effective diagnosis and monitoring of the disease [40]. The use of THz scanning systems as an objective assessment tool for DED has been explored in various studies and may provide a novel alternative to the existing diagnostic methods for DED [41,42].

In a series of studies led by Ozheredov et al. [41,42], which employed THz imaging to observe the corneal tear film in humans, THz reflectivity was identified as a measure of the dynamic status of the human corneal tear film. This was demonstrated in the study by instructing the participant to maintain his eyes open for extended periods prior to blinking, while continuously measuring the THz reflectivity of the eye using a non-contact method. The results showed a decrease in THz reflectivity over time before blinking, which returned to baseline levels after blinking. This observation supports the hypothesis that THz reflectivity correlates with the state of the corneal tear film, given that the physiological restoration of the corneal tear film is facilitated by blinking [41]. Further research established a linear fit for the gradient of the decay in THz reflectivity between blinks, following an examination of the eyes of 29 participants [42]. By utilizing the linear model to classify DED, a significant decrease in the THz reflectivity of the eyes between blinks was observed in dry eye patients [43]. This finding is in agreement with our observation, which also demonstrated a gradual decline in the THz intensity with time after eye blink (Figure 4). These studies confirmed the accelerated degradation of THz reflectivity during the inter-blink period in dry eye patients, highlighting the potential viability of THz imaging as an assessment tool for DED.

### 4.5. THz Imaging in Assessing Corneal Endothelium

Functioning as a non-regenerative monolayer of endothelial cells located on the posterior corneal surface, the corneal endothelium is one of five distinct sublayers of the cornea. The function of the corneal endothelium is two-fold: to facilitate diffusion of nutrients into the corneal stroma and to regulate stromal hydration. Passive diffusion of oxygen and metabolites from the aqueous humor is achieved through the intercellular tight junctions between endothelial cells, which acts as a porous barrier [43,44]. The regulation of stroma hydration is achieved through endothelial cells actively transporting excess fluid out of the stroma via a sodium–potassium pump [44,45]. The failure to sustain this function leads to the loss of corneal transparency [46,47]. The conventional technique for evaluating corneal endothelial cell health relies on specular microscopy. This non-invasive diagnostic approach captures high-resolution magnified images to examine the size, shape, density, and general state of the endothelial cells. Alternatively, the evaluation of CTWC fluctuations as a consequence of the compromised function of the corneal endothelium in clinical practice may render supplementary insights for early diagnosis of corneal edema. The proposed methods for evaluating CTWC include Confocal Raman spectroscopy and Brillouin microscopy. However, these techniques face limitations, such as the requirement for high fluence, which may pose challenges for potential in vivo applications [48,49].

THz spectroscopy presents as an alternative non-invasive imaging modality that has been explored in studies for its potential in measuring endothelial layer functionality. Utilizing terahertz time-domain spectroscopy (THz-TDS) within frequencies of 0.4 and 0.8 THz, significant differences in THz responses of ex vivo corneal samples with damaged versus healthy endothelial layers were demonstrated [50]. Differing degrees of endothelial layer integrity of the corneal samples were induced in the 19 porcine eye globe samples in the study through the application of varying magnitude of intraocular pressure of either 25, 35, or 45 mmHg for 4 h, followed by subsequent normalization to physiological pressures of 15 mmHg for the next 4 h. The differences in the mean THz spectral slopes were consistent with the variation in corneal endothelial cell density, which was evaluated via scanning electron microscopy imaging, with the more damaged samples exhibiting larger absolute THz spectral slope values than healthy samples [50]. There is a strong linear relationship between two THz spectral parameters and endothelial cell density. Using the mean THz spectral slopes of elevated and physiological intraocular pressure phases (SElev and SPhys) as predictors, a classification model based on a support vector machine algorithm was developed, which reported a receiver operating characteristics (ROC) area under the curve value of 0.91 ± 0.12 to diagnose an endothelial cell density <3000 cells/mm^2^. This study represents the first proof-of-concept demonstrating the viability of THz scanning systems as a conduit for evaluating the functional integrity of the corneal endothelium. Additionally, the integration of THz spectroscopy as an adjuvant tool alongside additional techniques, such as IVCM or specular microscopy, may offer valuable insights into discerning the sensitivity and magnitude of corneal tissue response to endothelial damage in an in vivo context.

### 4.6. THz Imaging in Corneal Hydration

Corneal hydration plays a vital factor in maintaining overall ocular health and vision. The delicate balance of water composition within the cornea is crucial in preserving corneal transparency and disruptions in the hydrodynamics of the cornea result in visual perturbations. If excessive water accumulates within the cornea, this leads to the development of corneal edema, which constitutes one of the major pathological causes of corneal opacity. The clinical implications of advanced and irreversible corneal edema are severe, with the sole treatment option being corneal transplantation. This intervention, however, presents additional challenges, such as the global scarcity of human donor corneas and the necessity for lifelong topical immunosuppressants for transplant recipients [51]. This highlights that the monitoring of corneal hydration is crucial to avert progression to end-stage deterioration [31,33]. Moreover, corneal hydration has been identified as a risk factor that influences both the efficacy and safety outcomes of refractive surgery [28,52]. No current techniques remain to accurately assess corneal hydration. However, surrogate measures involve assessing central corneal thickness (CCT) using an ultrasonic pachymeter, anterior segment optical coherence tomography (ASOCT), and corneal tomography to indirectly evaluate corneal hydration [52,53]. Nevertheless, these surrogate measurements do not reflect the absolute hydration values. The use of THz technology has been proposed as a potential modality for corneal hydration assessment.

The use of THz technology to evaluate corneal hydration has been explored in both in vivo and ex vivo animal studies [32,54,55,56,57,58]. One of the earlier ex vivo experiments assessing corneal hydration utilized nine porcine corneal samples that were left to soak in polyethylene glycol solutions of differing concentrations for three days to achieve equilibrium water concentrations in the cornea [56]. Corneal THz reflectivity and corneal water concentration, determined by mass, were measured using THz spectroscopy and an automated weighing scale, respectively [56]. The study demonstrated a clear decrease in THz-reflected intensity in the corneas with reduced absorbed water content, supporting the viewpoint of the potential value of THz reflectivity in quantifying corneal hydration. Our group’s ex vivo research further showed that the assessment of hydration levels in the corneal stromal layer through THz spectroscopy correlated with the corresponding CCT of the samples, as measured by ASOCT [31]. This linear relationship between corneal THz reflectivity and corneal water concentration has also been supported by various earlier and subsequent ex vivo studies [54,56,57].

More recently, our group published an in vivo animal study in which THz waves of frequencies up to 3 THz were applied to eight rabbit corneas to evaluate the effectiveness of THz technology in evaluating corneal edema [32]. Varying severity of corneal edema was induced using a Descemet membrane stripping model, with the THz and CCT measurements documented over the following 10 days after surgery [59]. By measuring the THz reflection signal intensity and CCT via ASOCT scans, THz reflection signal intensity correlated with the corresponding corneal thickness in a linear fashion. In a similar vein, Iomdina et al. [55] conducted an in vivo rabbit model study where 16 eyes of 8 rabbits were exposed to B-band ultraviolet-induced damage and conducted a series of examinations, including the THz reflectivity, of rabbit cornea for 30 days. The corneal reflectivity coefficient corresponded with the progression of corneal edema, as evidenced by corneal thickening [55]. Additionally, the hydration sensitivity of THz scanning has been shown to be three parts per thousand (ppt), exceeding the benchmark of one ppt, indicating its capability to distinguish between diseased and healthy corneal tissues in the context of corneal hydration [58]. These support the viewpoint of the potential application of THz scanning systems for corneal hydration for monitoring disease progression and treatment efficacy.

### 4.7. Corneal Elasticity

The cornea is considered a viscoelastic tissue with complex biomechanical attributes, including non-linear elasticity, anisotropy, and viscoelasticity. These biomechanical properties serve a crucial role in the maintenance of the normal structure and function of the cornea [60]. The mechanical resistance of the human cornea can be altered by iatrogenic factors, including refractive and therapeutic treatments, as well as ocular or systemic diseases [61]. Alterations in biochemical properties due to surgery or pathological diseases can introduce corneal swelling, affecting visual acuity [62]. Corneal elasticity evaluation has attracted increasing interest in clinical practice, particularly for its potential role in predicting the cornea’s response to surgical and therapeutic interventions or the progression of keratoectasia, given that biochemical changes often precede clinical symptoms [63]. Current methods for assessing corneal elasticity include Brillouin microscopy and the assessment of shear wave propagation. The former is an advanced non-invasive imaging technique that measures corneal biomechanical properties by analyzing the natural frequency shift of light upon the interaction between photons and acoustic photons. The frequency change can be utilized to produce an estimate of the viscoelasticity of the cornea but with limitations such as lengthy measurement times and inconsistent accuracies [64]. The latter has been used to provide an estimate of the corneal elasticity [65,66,67]. Current commercial options include the Corvis ST tonometer (OCULUS Optikgeräte GmbH, Wetzlar, Germany), which employs an ultra-high-speed Scheimpflug camera to quantitatively measure corneal biomechanical properties during a rapid air-puff application [68]. Limitations of air puff tonometers like the Corvis ST include the influence of factors like intraocular pressure on its measurements [69] and the need for averaging multiple measurements to achieve reliable accuracy [70,71]. Other proposed techniques to assess corneal elasticity include optical coherence elastography and high-resolution ultrasound strain imaging of the cross-sectional cornea using an ultrasound elasticity microscope [72,73]. Whilst not restricted in clinical use, these methods continue to face some enduring criticisms, including their time-consuming nature and the accuracy of their measurements. The utility of THz technology, as a non-contact and non-invasive detection method, has been considered as a potential alternative option to bolster the currently available tools in evaluating corneal biomechanical properties.

THz spectroscopy for evaluating corneal elasticity has been studied through an ex vivo investigation involving human cadaveric corneas [34]. By utilizing the THz spectroscopy as a probing tool to induce light pressure to produce mechanical corneal deformation, the changes in THz signal due to the light pressure correlated to the elastic property of the corneas. This was evidenced by using the relationship between the THz signal phase shift and the refractive index shift to determine each cornea’s Young modulus, the key mechanical characterization parameter indicating the shapes of the anterior and posterior cornea. The calculated values were consistent within the range of corneal Young’s modulus values reported in the literature [74,75,76]. Additionally, other imaging techniques such as atomic force microscopy (AFM) have also found success in measuring corneal elasticity, reporting quantitative measurements of individual collagen fibrils and the subsequent derivation of the elastic modulus values in human corneas [77,78]. Similar to THz spectroscopy, the mean elastic modulus determined via AFM was found to align with the range of corneal Young’s modulus values within the literature, providing a validating tool for the viability of THz technology in measuring corneal elasticity. While corneal tissue behavior remains considerably more complex in the actual in vivo biological environment, the successful demonstration of THz technology to ex vivo corneal tissue supports the perspective that THz technology could serve as a promising future non-invasive method for subsequent in vivo studies and clinical evaluation of corneal parameters biomechanics. The potential clinical application of THz technology for assessing corneal biomechanics may have significant impacts on the clinical approach to corneal diseases, such as keratoconus, since this is fundamentally driven by the biomechanical weakening of the cornea [79,80,81]. Greater insight into and evaluation of corneal biomechanics through these novel modalities such as THz technology will empower clinicians to objectively assess the altered biomechanical behavior of the cornea, both at the point of diagnosis and post-treatment care, positioning it as a potential tool to change the landscape of corneal disease diagnosis and treatment.

### 4.8. Corneal Scars

The potential utility of THz technology has been explored in an ex vivo rabbit corneal scar model via an experimental setup [33]. Corneal scars were induced at the center of the corneas of four rabbit eye models through irregular phototherapeutic keratectomy, with the samples left for scar tissue development over six to eight weeks prior to evaluation. THz spectroscopy and imaging demonstrated sufficient contrast for evaluating corneal density through THz absorption coefficient spectra, as evidenced by additional absorption peaks when compared to control corneas. Furthermore, THz refractive index spectra pinpointed the corneal scars, confirmed via slit lamp images, by exhibiting diminished refractive index at the scar sites, in contrast to the unaffected peripheral corneal tissue. In our previous study, corneal imaging on corneal scars was conducted utilizing a raster scan technique, which involves systematically capturing images in a grid-like pattern. This process was enhanced through the automatic movement of a three-axis stage, allowing for precise positioning and alignment of the imaging system. This approach enables the visualization of the corneal structure, facilitating detailed analysis and assessment of the density and size of the corneal scars (Figure 5). THz technology’s ability to illustrate the scar tissue profile with depth variation and density underscores its advantages over ASOCT, which relies on hyperspectral methods in the visible and near-infrared regions, in mapping the spatial distribution and compositional alterations of corneal scars. Current limitations of THz technology in assessing corneal scarring may arise particularly in faint or small scars, where the scar composition may not differ sufficiently for clear differentiation. Nevertheless, recent evidence continues to affirm the potential of THz technology as an emerging quantitative imaging tool for the assessment of corneal scarring.

The applications of THz technology in corneas and corneal diseases are summarized in Table 1 and Figure 6.

## 5. Future Directions

Although the field of THz technology has demonstrated great potential in disease diagnosis and monitoring in corneal studies with its unique physicochemical properties, making it a promising candidate for biochemical applications, particularly in assessing dry eye, corneal scarring, and hydration, additional efforts are needed to fully understand and establish its role alongside its potential long-term effects before it can be broadly applied in clinical contexts.

A significant body of recent studies explored in this article is predominantly focused on animal models in ex vivo settings. These studies have deepened the understanding of its potential as a diagnostic tool for corneal diseases, showcasing distinct advantages over existing approaches to corneal health assessment. Subsequent research directions could focus on validating these findings using human in vivo corneal studies, thereby achieving a closer approximation to in vivo physiology. Clinical trials are ongoing by the authors’ research group to determine the diagnostic accuracy and long-term safety of THz technology in the context of assessing corneal diseases in real-world conditions. Further clinical assessments of the THz technology remain a paramount step in validating its real-world viability, yet we acknowledge this as an inherent obstacle in integrating novel modalities into clinical practice. Like other new imaging modalities, the success of THz technology in real-world contexts may also be constrained by real-world challenges and limitations, including the need to compete with established best practices and cost constraints. Additionally, improving current THz imaging profiles or integrating existing imaging modality may further enhance the precision of THz technology in clinical applications, such as in the detection of subclinical corneal edema, real-time monitoring of the dynamics of tear film or corneal hydration changes during excimer laser-based refractive surgery, and superficial or faint corneal scars. To achieve this, sustained and close collaboration between medical specialists and bioengineering experts is instrumental to achieving the successful delivery of biomedical applications of THz technology in the landscape of clinical medicine. Given the potential for THz technology to be integrated with advanced computational technologies such as artificial intelligence and deep and machine learning, further studies could enhance its diagnostic capabilities for supporting clinical decision-making [82]. By covering these research trajectories, the field can achieve a deeper understanding of diagnostic accuracy and its practicality, which are essential for the development and acceleration of the practical application of THz imaging systems in the context of corneal health assessment and treatment.

## 6. Conclusions

This systematic review provides a comprehensive overview of THz radiation’s fundamental characteristics, its safety profile, and its potential biomedical applications in the context of the cornea and ocular surface. A range of studies has highlighted the applicability of THz imaging systems in evaluating different facets of corneal pathologies, including corneal hydration, elasticity, scarring, endothelial damage, and DED. We propose that the most promising initial applications of THz technology in corneal diseases would be in the assessment of corneal hydration and DED. This assertion is supported by the property of high sensitivity of THz technology to water, making it especially promising for assessing corneal pathologies associated with water content abnormalities, such as corneal edema and DED. The growing availability of preclinical trial data on these conditions and their limited options for objective clinical assessment further support this notion. The utilization of THz systems has grown increasingly widespread across interdisciplinary fields and looks poised to drive further growth in a myriad of disciplines beyond medicine. Within the field of medicine, THz systems have notably been developed into various configurations to analyze samples that range from molecules to cells and tissues. The potential application of these systems encompasses a wide spectrum of areas, including fundamental physical and chemical investigations to pathological diagnosis. Although significant strides are still needed, we envisage that continued research and technological advancement in THz technology will progressively bridge the gap between research and its practical clinical application.

## Figures and Tables

**Figure 1 bioengineering-12-00045-f001:**
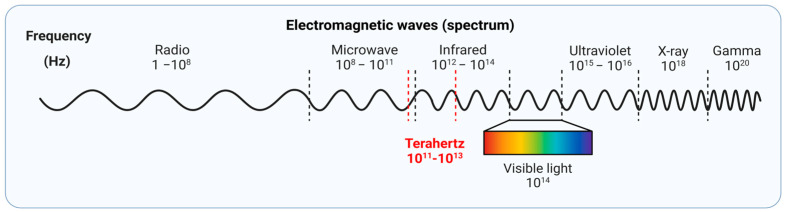
Illustration of Terahertz Radiation in the electromagnetic spectrum. The figure was created with Biorender.com.

**Figure 2 bioengineering-12-00045-f002:**
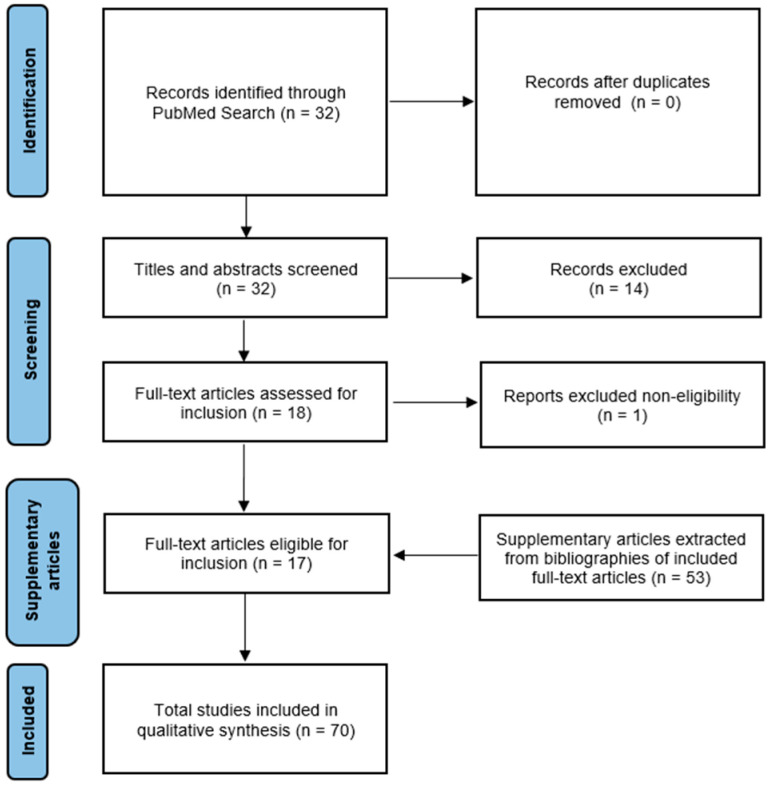
Flow diagram of the literature selection process for the present article.

**Figure 3 bioengineering-12-00045-f003:**
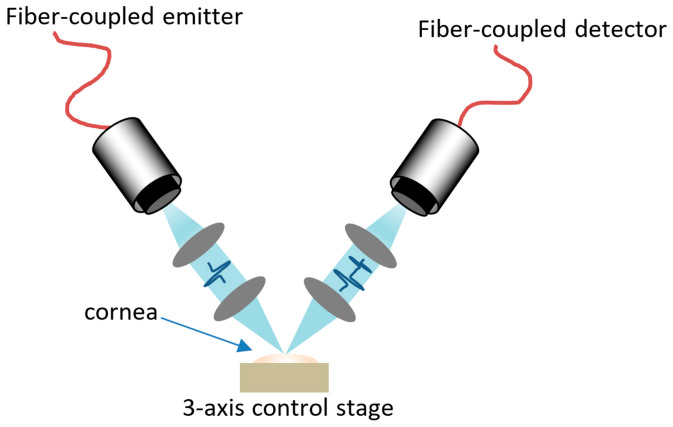
The schematic diagram of the THz system used in cornea application.

**Figure 4 bioengineering-12-00045-f004:**
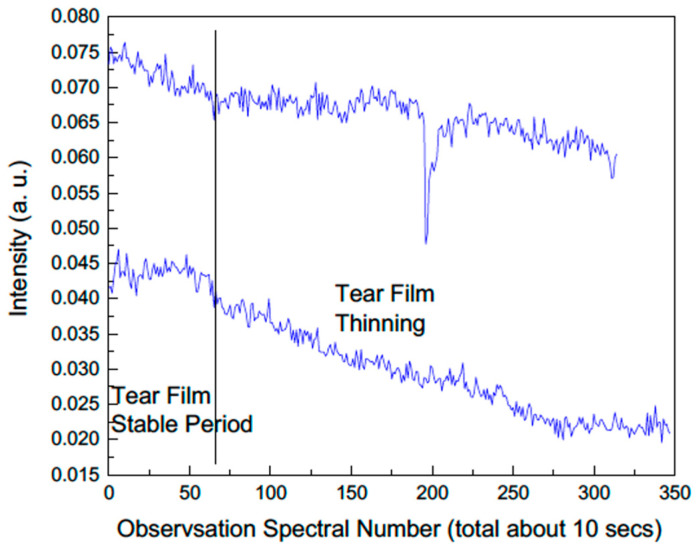
The changes in THz total intensity with time in two patients with DED (unpublished data from the authors).

**Figure 5 bioengineering-12-00045-f005:**
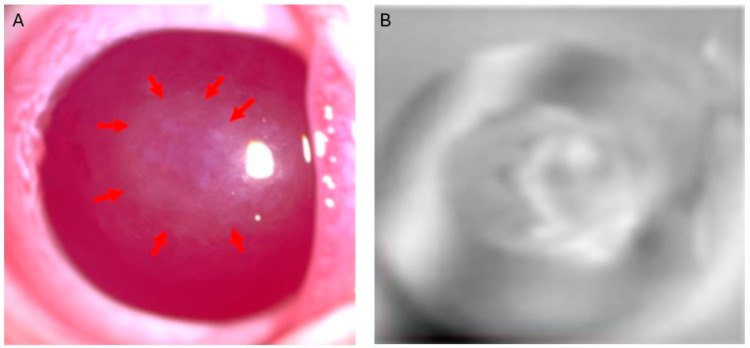
Illustration of THz spectroscopy and imaging of corneal scars in rabbits. (**A**) Slit lamp photos of a laser-induced scar (arrows). (**B**) THz 2 2D B-scan reconstructed images delineated the scar and demonstrated the scar density (unpublished data from the authors).

**Figure 6 bioengineering-12-00045-f006:**
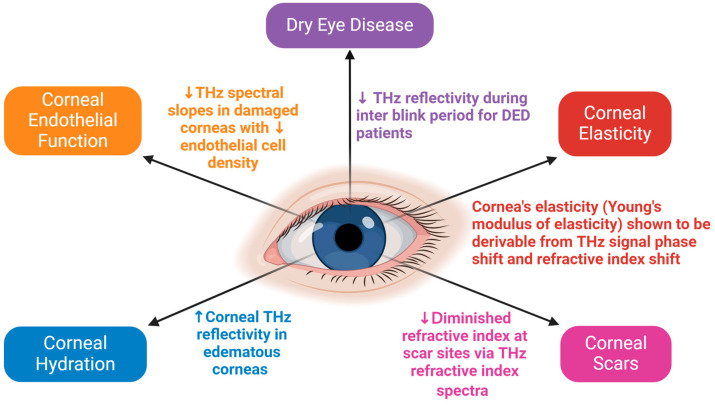
Illustration of the potential clinical applications of THz technology in assessing corneal diseases. The figure was created with BioRender.com.

**Table 1 bioengineering-12-00045-t001:** The application of Thz technology for ocular surface and corneal pathologies.

Indication	Study Design	Findings
Dry eye Disease	Clinical in vivo human THz imaging studies (*n* = 1 patient) [41] and (*n* = 29 patients) [42]	• Decrease in THz reflectivity over time before blinking, which returned to baseline levels after blinking [41]. • Significant decrease in THz reflectivity in the inter-blink period for DED patients [42]
Corneal endothelium functionality	Ex vivo porcine model study utilising THz-TDS spectroscopy [1]	• Significant linear correlation between THz spectral parameters and endothelial cell density [50]. • Area Under Curve value of 0.91 ± 0.12 of a classification model utilising mean THz spectral slopes to diagnose an endothelial cell density <3000 cells/mm^2^ [50].
Corneal Hydration	Ex vivo rabbit [54] and porcine THz imaging studies [56,57]	• Significant increase in corneal THz reflectivity with increasing corneal water concentration [54,56]. • Reduction in THz reflectance from the centre of cornea to edges, which aligns with the higher hydration levels in the corneal center [57]. • Strong linear correlation in THz reflection signal intensity with corresponding CCT measurements [31,32]. • Changes in the corneal reflectivity coefficient corresponded to corneal edema progression [55]. • Hydration sensitivity of THz scanning reported to be 3 ppt suggests sufficiently powered to highlight corneal hydration abnormalities [58].
	In vivo rabbit THz imaging studies [32,55,58]	
Corneal elasticity	Ex vivo human cadaveric corneas using THz spectroscopy (*n* = 6 corneas) [34]	• THz signal phase shift and refractive index shift were used to determine the Young’s modulus of elasticity for each cornea, and the values were consistent with the standard range for corneal Young’s modulus [34].
Corneal scars	Ex vivo rabbit THz imaging and THz spectroscopy [33]	• Significant decrease in refractive index of scar centres compared to control corneal samples, observed via THz refractive index spectra [33]. • Significantly increased THz reflection peak intensity in the scar samples, compared to controls, observed via THz-TDS [33]. • THz imaging contrast sufficiently outlined the laser-induced corneal scars, highlighting its ability to assess corneal scar size [33].

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
