# Peer review of "The Application of Terahertz Technology in Corneas and Corneal Diseases: A Systematic Review"

_bioengineering, 2025, doi:10.3390/bioengineering12010045_

Round 1
Reviewer 1 Report
Comments and Suggestions for Authors
Very nice paper - an extensive narrative review by an experienced group of authors in this field.
My only suggestion - which the authors have already partially addressed - is to discuss the implications that virtually all of the studies cited in this paper consist of proof of concept, very early stage research, not even Phase 1 studies. Consequently there is no basis other than very preliminary, pre-clinical research to suggest the potential applications of any of this in the real world of medicine.
Authors would provide a service by stressing the importance and difficulty of clinical assessments of a proposed THz technique, and the very chancy likelihood of success in the real world of medicine, where new methods have to compete with existing best practices and severe cost constraints.
Would the authors comment: where is the "low hanging fruit" of THz technology likely to be for successful clinical applications?
Readers should be advised of the need to collaborate with medical specialists early on to avoid investing a lot of engineering work on applications that have little chance of succeeding in medical practice. I have seen it happen many times.
This is important because the biomedical engineering literature is filled with papers reporting early stage research on biomedical applications of microwaves but few if any proposed microwave applications have gained any measure of success in clinical medicine.
- Just suggestions for the authors to take or not.
Author Response
Point-by-Point Response to Reviewer 1
Reviewer 1’s Comments:
Very nice paper - an extensive narrative review by an experienced group of authors in this field.
Comment 1: My only suggestion - which the authors have already partially addressed - is to discuss the implications that virtually all of the studies cited in this paper consist of proof of concept, very early stage research, not even Phase 1 studies. Consequently there is no basis other than very preliminary, pre-clinical research to suggest the potential applications of any of this in the real world of medicine. Authors would provide a service by stressing the importance and difficulty of clinical assessments of a proposed THz technique, and the very chancy likelihood of success in the real world of medicine, where new methods have to compete with existing best practices and severe cost constraints.
Authors Response 1: We thank you for your comments and suggestions regarding our paper. We agree with your comment on the importance and difficulty of clinical assessments of a novel imaging modality as well as the potential real-world challenges one may face. We have now reflected it in the revised manuscript:
“Further clinical assessments of the THz technology remain a paramount step in validating its real-world viability, yet we acknowledge this as an inherent obstacle in integrating novel modalities into clinical practice. Like other new imaging modalities, the success of THz technology in real-world contexts may be constrained by real-world challenges and limitations, including the need to compete with established best practices and cost constraints.” (Page 12. Line 438-443)
Comment 2: Would the authors comment: where is the "low hanging fruit" of THz technology likely to be for successful clinical applications?
Authors Response 2: Thank you for your comment. Our group has been working on THz technology for the past 10 years, from proof-of-concept studies, to preclinical, phase I and phase IIa studies. From our work and data, the assessment of corneal hydration and dry eye disease will be likely to be for successful initial clinical applications of THz technology in corneal diseases. This is due to the greater availability of data from pre-clinical trials on these conditions, combined with the increased sensitivity of THz technology to water, which suggests that corneal pathologies associated to water content abnormalities such as the mentioned are better positioned for successful clinical translation and application. Furthermore, the current clinical methods to objectively evaluate corneal hydration status remains limited at present, making THz technology as an objective tool a more attractive prospect. Our group is currently conducting a phase II clinical trial on the application of THz imaging system in patients with corneal edema or dry eye.
We have revised the manuscript to include this point: “We propose that the most promising initial applications of THz technology in corneal diseases would be in the assessment of corneal hydration and DED. This assertion is supported by the property of high sensitivity of THz technology to water, making it especially promising for assessing corneal pathologies associated to water content abnormalities, such as corneal edema and DEDThe growing availability of preclinical trial data on these conditions and their limited options for objective clinical assessment further support this notion.” (Page 12, Line 462-468)
Comment 3: Readers should be advised of the need to collaborate with medical specialists early on to avoid investing a lot of engineering work on applications that have little chance of succeeding in medical practice. I have seen it happen many times. This is important because the biomedical engineering literature is filled with papers reporting early stage research on biomedical applications of microwaves but few if any proposed microwave applications have gained any measure of success in clinical medicine.
Authors Response 3: We are thankful for your comment, and we agree that multi-disciplinary collaborations are indeed important for the translation and clinical adoption of biomedical engineering technology. We have revised the manuscript to reflect this in the ‘Future Directions’ section:
"To achieve this, sustained and close multi-disciplinary collaborations between clinicians, bioengineering and bioimaging experts is instrumental to achieve the successful translation of THz technology in the landscape of clinical medicine.” (Page 12, Line 447-450)
Reviewer 2 Report
Comments and Suggestions for Authors
Dear Author,
This manuscript is well-written and supported by significant research papers. While terahertz technology holds considerable promise for biomedical research, the long-term effects—extending beyond a week or even months—should also be explored or incorporated into the discussion if relevant reports are available. Authors should highlight or include research papers which further validated terahertz technology data at molecular events.
Additionally, it would be beneficial to assess corneal elasticity using atomic force microscopy and validate the findings with a larger sample size to strengthen the study's conclusions.
Finally, the manuscript would be further enhanced by highlighting the role of artificial intelligence (AI) in corneal disease diagnosis and treatment, as well as exploring its correlation with terahertz technology.
Author Response
Point-by-Point Responses for Reviewer 2
Reviewer 2:
Comment 1: While terahertz technology holds considerable promise for biomedical research, the long-term effects—extending beyond a week or even months—should also be explored or incorporated into the discussion if relevant reports are available. Authors should highlight or include research papers which further validated terahertz technology data at molecular events.
Authors Response 1: We thank the reviewer for the comment. In our review, we have included available published literature reporting on the reported safety profile of THz technology on the corneas [29,34,35]. However, there remains a paucity of data regarding long-term effects of THz technology to our knowledge, which is a limitation we acknowledged in the original manuscript. We now have expanded this in the Future Directions section of the revised manuscript: “Additional efforts are needed to fully understand and establish its role alongside its potential long-term effects before it can be broadly applied in clinical contexts.” (Page 12, Line 427-429)
Comment 2: Additionally, it would be beneficial to assess corneal elasticity using atomic force microscopy and validate the findings with a larger sample size to strengthen the study's conclusions.
Authors Response 2: Thank you for the comment. We agree that accessing corneal elasticity using atomic force microscopy would help validate the findings from THz research. We now have added this to the corneal elasticity subsection of the revised manuscript to strengthen the study’s conclusion. We have also included 2 corresponding references to the manuscript (References 76 and 77).
“Additionally, other imaging techniques such as atomic force microscopy (AFM) has also found success in measuring corneal elasticity, reporting quantitative measurements of individual collagen fibrils and the subsequent derivation of the elastic modulus values in human corneas [76,77]. Similar to THz spectroscopy, the mean elastic modulus determined via AFM were found to align with the range of corneal Young’s modulus values within the literature, providing a validating tool for the viability of THz technology in measuring corneal elasticity. (Page 9, Line 371-377)
“Meek, K.M.; Fullwood, N.J. Corneal and Scleral Collagens—a Microscopist’s Perspective. Micron 2001, 32, 261–272, doi:10.1016/S0968-4328(00)00041-X.”
“Last, J.A.; Russell, P.; Nealey, P.F.; Murphy, C.J. The Applications of Atomic Force Microscopy to Vision Science. Invest Ophthalmol Vis Sci 2010, 51, 6083–6094, doi:10.1167/iovs.10-5470.”
Comment 3: Finally, the manuscript would be further enhanced by highlighting the role of artificial intelligence (AI) in corneal disease diagnosis and treatment, as well as exploring its correlation with terahertz technology.
Authors Response 3: We are thankful for your suggestion. We agree about the potential value of artificial intelligence in relation to THz technology in corneal disease diagnosis and treatment and have revised the manuscript accordingly and included a corresponding reference. (Reference 79)
“Given the potential for THz technology to be integrated with advanced computational technologies such as artificial intelligence and machine learning, further studies could enhance its diagnostic capabilities for supporting clinical decision-making.” (Page 12, Line 450-453)
“Gezimati, M.; Singh, G. Terahertz Data Extraction and Analysis Based on Deep Learning Techniques for Emerging Applications. IEEE Access 2024, 12, 21174–21198, doi:10.1109/ACCESS.2024.3360930.”